# Reverse Osmosis Treatment of Wastewater for Reuse as Process Water—A Case Study

**DOI:** 10.3390/membranes11120976

**Published:** 2021-12-10

**Authors:** Marjana Simonič

**Affiliations:** Faculty of Chemistry and Chemical Engineering, University of Maribor, 2000 Maribor, Slovenia; marjana.simonic@um.si

**Keywords:** alumina production, reverse osmosis, fouling, wastewater reuse

## Abstract

The aim of this work was to purify mixed wastewater from three different production processes in such a manner that they could be reused as process water. The maximum allowed concentrations (MAC) from the Environmental Standards for emissions of substances released into surface water were set as target concentrations. Wastewaters contained solid particles, sodium, aluminium, chloride, and nitrogen in high amounts. Quantitatively, most wastewaters were generated in the production line of alumina washing. The second type of wastewater was generated from the production line of boehmite. The third type of wastewater was from regeneration of ion exchangers, which are applied for feed boiler water treatment. The initial treatment step of wastewater mixture was neutralisation, using 35% HCl. The precoat filtration followed, and the level of suspended solids was reduced from 320 mg/L to only 9 mg/L. The concentrations of ions, such as aluminium, sodium and chlorides remained above the MAC. Therefore, laboratory reverse osmosis was applied to remove the listed pollutants from the water. We succeeded in removal of all the pollutants. The concentration of aluminium decreased below 3 mg/L, the sodium to 145 mg/L and chlorides to 193 mg/L. The concentration of nitrate nitrogen decreased below 20 mg/L.

## 1. Introduction

Alumina (Al_2_O_3_) is considered a basic catalytic material support due to its good mechanical properties, such as high strength, chemical and physical stability, additionally, its high thermal resistance, and thermal conductivity [1]. Boehmite (AlOOH) is used as the raw material for the preparation of alpha and gamma-alumina phases whose properties such as morphology, specific surface area, and porosity, depend strongly on the boehmite structure [2].

The disposal of washing water into the environment is not allowed without treatment, because some parameters, such as Al, Na and Cl, exceed the maximum allowed values [1]. Water impurities concentrate during boiler operation. Boiler water must be softened properly before use. However, without periodic water removal (blowdown), problems such as scale deposits, corrosion and embrittlement may occur [3]. A study was reported, conducted on the utilisation of a simulated boiler blowdown for incorporation into cement-based materials. The results indicate that the use of waste brine in cemented backfill applications is feasible. The phosphate addition could result in the formation of deposits such as iron phosphate. Phosphate corrosion is assumed as a significant concern in phosphate treated steam boilers [4]. Basic feedwater treatment involves ion exchange. For existing industries operating at a low scale, resin-assisted separation continues to offer an attractive option [5]. After regeneration of an ion exchanger, acidic and alkaline effluents are generated, containing chlorides, sulphates, nitrates, silicates, etc.

Reuse of wastewater is recognised in most water-scarce countries [6]. Instead of discharging industrial effluents into rivers and streams, reverse osmosis (RO) membranes can be used to treat the wastewater and reuse it as process water in companies. Spiral wound modules (SWMs) are most widely adopted among the commercially available RO membrane modules [7]. The analysis of the design of spiral wound modules and correlation of experimental and model values can help to identify the module geometry and spacer design for specific applications [8]. The permeability coefficients with the feed spacer design have a relevant impact on the performance of RO membranes in terms of production, permeate quality and specific energy consumption [9]. This is due to the relationship between the feed spacer geometry and pressure drop and concentration polarisation phenomena. The operation of an RO system deals with the impact of fouling, [10] which is the main concern of this technology and along with optimal operational conditions the main thing responsible for losing performance and efficiency in a long term-operation [11]. Spiral wound modules offer ease of operation, fouling control, and a high permeation rate and packing density. Applications of spiral wound modules include desalination, water treatment, water reclamation and treatment of industrial wastewater. A water recycling system has been documented using RO technology for the treatment of oleochemical wastewater [12]. The fouled RO membrane required a high operating pressure in order to obtain a consistent permeate flow. In general, RO membranes are subjected to surface fouling and scaling, which can pose a significant problem when reverse osmosis is used [13]. Scaling is one of the limiting factors for increasing the flux recovery rate and improving the efficiency of the process [14]. It is well known that filtration prior to RO is essential for maintaining efficiency and protecting the membranes` functions. The use of RO membranes largely reduces the chemical consumption when membrane separation is coupled with chemical treatment, due to the capability of the RO membranes of removing dissolved solids [12]. The application of the RO process compared to other conventional thermal technologies for desalination of brackish water has increased remarkably, due to its high purification efficiency at low cost, and low energy consumption [15].

The main objective of the present study was to investigate the possibilities of recycling mixed wastewater from three different processes: The first after alumina washing, the second after boehmite production and a third from the regeneration of ion-exchangers for feed boiler water treatment. The chemical analyses of individual wastewaters did not differ much, therefore, wastewater from all three lines was collected in a feed tank, neutralised, and filtered using precoat filters. This was followed by RO treatment using two selected membranes, FILMEC XLE (Dow) and ESPA (Nitto), chosen from among the major membrane manufacturers. Treatment was tested in two membrane modes: With and without concentrate recycling. The quality of the RO permeate after both modes was compared with the Slovenian Regulations for water discharge into the environment. The organic content was determined, expressed as chemical oxygen demand (COD).

## 2. Materials and Methods

Alumina washing waters are collected in a feed reservoir. The alumina is firstly washed in an HCl acid solution and then filtered. The solid particles and the solution are separated. The solution contains large amounts of inorganic compounds, and must be treated before it is released into the environment. The second wastewater is generated after boehmite washing. Boehmite is aluminium oxide hydroxide that forms from Al(OH)_3_. The lattice spacing in the material is 0.117 nm. The structure [2,16] as well as hydrophobicity, light transmittance, thermal stability, and mechanical durability [17] has been characterised extensively in the literature. Wastewater flows into the decanter, and it is then mixed with the washing water from the alumina. The third type of wastewater is generated after regeneration of an ion exchanger for boiler water treatment.

The amount of water from alumina washing is 72,000 m^3^/y, from boehmite 14,500 m^3^/y and from boiler water 14,400 m^3^/y. In such ratios (5:1:1) wastewater was mixed in the 100 m^3^ feed tank in the laboratory.

The feed solution after precoat perlite filtration flowed from the tank through the valve and pump to a Culligan Aqua-Cleer RO system. The system was equipped with volume counters (2) and pressure gauges (3), as seen from Figure 1. The system allows transmembrane pressure (*TMP*) up to 15 bar. The sample flows through the 80 µm filter and 5 µm filter for removal of the bigger particles. Such pre-treated solution is gathered in a feed tank (4). The permeate and retentate were withdrawn continuously from the system in the initial four experiments. In the final experiment the permeate and concentrate were recycled into the feed tank. The capacity was 1.5 L/h for the permeate and 1.9 L/h for the retentate at room temperature.

The feed samples and permeate were taken at the end of the trial and subsequently analysed. The wastewater flux was measured after every repeated trial with mixed wastewater.

Dow-FILMTEC XLE (DuPont-Filmtec, Wiesbaden, Germany) and ESPA-2521 (Nitto, Nottinghamshire, UK) RO membranes were used with the properties listed in Table 1. The membrane module dimensions were 0.287 m in length and 0.117 m in diameter.

The analyses of the feed wastewater and permeate were performed according to ISO Standards in three replicates. The Standard methods are summarised in Table 2. The analyses were chosen in accordance with the Slovene legislation for wastewater emission into the environment [18]. Important parameters among the general parameters are suspended solids mass concentration and pH. Absorbance at 436 nm was measured as an indication of inorganic contamination. The sum of all organic compounds in the water samples was determined as *COD*. Inorganic species are measured as chloride, sulphate and nitrate ions using Ion chromatography (IC). Metals (Na, Cu, Cr, Ti, Zn, Ni, Mg), including Al, were determined by inductively coupled plasma and mass detector (ICP-MS). SiO_2_ was determined spectrophotometrically. Suspended solids (SS) were analysed with an Imhoff funnel.

### 2.1. Calculations

Rejection *R* was determined according to Equation (1):*R* = (*c*_f_ − *c**_p_*)/*c*_f_(1)

Concentration factor *f*_c_ was determined according to Equation (2):*f*_c_ = 1/(1 − *q*)(2)
where

*c*_f_ = concentration in the feed solution (mg/L/)

*c*_p_ = concentration in the permeate solution (mg/L/)

*q* = water flow (L/h)

One of the most common methods of determination of fouling is the membrane filtration index (MFI) [19]. Cake layer formation is proposed as the dominant mechanism. The MFI test is performed by filtration of water through a 0.45 um filter with constant pressure in dead-end mode. The MFI can be calculated as seen from Equation (3) [19]:(3)$$\frac{t}{V}=\frac{µ\;R_{m}}{TMP.A}+\frac{µ\;\;I}{2\;TMP\;A^{2}}\;V$$

By plotting *t/V* versus *V* (permeate volume) the MFI is defined as the slope of a straight line after the initial linear section. *TMP* represents transmembrane pressure (bar), *A* is the membrane area (m^2^), *R*_m_ is the membrane resistance (m^−1^), *µ* is viscosity (Pa. s) and *t* is the filtration time (s). The higher the fouling potential for a given solution, the higher will be the MFI value.

### 2.2. SEM Imaging

The washing solutions after membrane cleaning were examined with an FEI, SIRION-400 field emission scanning electron microscope (FESEM).

## 3. Results and Discussion

### 3.1. Physico-Chemical Analyses

After filtration of all samples, the parameters listed in Table 3 were measured in the wastewater samples. Wastewater was taken from the washing process line 10 times and mixed. Such average sample was analysed for all parameters according to the Slovenian Regulations. Washing water in alumina production is denoted in Table 3 as WW1, washing water in boehmite production is denoted in Table 3 as WW2, and water after regeneration during feed boiling water condition as WW3. The results are gathered in Table 3. Al exceeded 3 mg/L in WW1. The suspended solids’ mass concentration was too high in the washing water in boehmite production (WW2), sodium was above the MAC and aluminium was very high, measured at 33 mg/L. In WW3 phosphorus was just above the MAC and the *pH* value was too alkaline. Absorbance at 436 nm exceeded the MAC for all samples. MAC represents the maximum allowed concentration according to the Slovenian Regulations [18]

### 3.2. Pretreatment with Precoat Filtration

The mixture of wastewater streams was neutralised to pH = 7 using HCl 35% (weight percents). An aliquot of 45 L of the mixture was vacuum filtered through 2 cm of perlite layer (200 g). The emphasis was on suspended solids` removal. The removal was 96%, which means that the measured concentration decreased down to 11 mg/L. The results are in good agreement with the reported study showing a 95% reduction in turbidity [20]. The concentration was acceptable for further treatment. Settleable solids decreased below 0.1 mg/L and were not problematic. Among metals Al was measured, and the removal was 64%, which was still above 3 mg/L. Other parameters from Table 3 were below the MAC.

During the second trial the same aliquot of mixture was filtered through 350 g of a 2.8 cm perlite layer. The results were similar compared with the first trial, but the filtration time was prolonged by some 15 min. The suspended solids’ removal was 97%, which means that the measured concentration decreased down to 9 mg/L. It was acceptable for further treatment. Settleable solids decreased below 0.1 mg/L, and were not problematic. Among metals Al was measured, and the removal was 66%, which was still above 3 mg/L. Other parameters were below the MAC.

According to the results and the literature [21] it could be concluded that increasing the amount of precoat did not improve the regeneration result.

### 3.3. Filtration with a Reverse Osmosis Membrane

The linear plot for permeability with Millipore water was determined using both RO membranes, FILMTEC XLE-2521 (XLE) and ESPA-2521 (ESPA), as shown in Figure 2. Similar values were also obtained in another study [22]. The permeability of both membranes was followed in the range from 6 bar to 10 bar. A small deviation was observed at 7 bars with both membranes, indicating that the highest flux was reached at 7 bar and 10 bar *TMP*. The permeate flux remained very similar to that of the trial at 10 bars with concentrate withdrawal. The wastewater fluxes were determined at 111 L/(m^2^.h) for XLE and 112 L/(m^2^.h) for ESPA at 10 bar. At other *TMP,* the flux was slightly lower compared to that at *TMP* = 7 bar and *TMP* = 10 bar. Therefore, 7 bar and 10 bar *TMP* were chosen for further trials with both membranes. The optimal *TMP* between 7 and 8 was reported in the literature [22,23].

The fluxes of mixed wastewater are shown in Figure 3 for both membranes at two different *TMP*; 7 bar and 10 bar were chosen according to the millipore water flux and the literature [22]. The fluxes of both membranes were the same at 7 bar, and the curves overlapped each other. Slightly higher fluxes were achieved at 10 bar with ESPA compared with the XLE membrane. The same trends were observed with the time dependence: During 30 min the flux decreased, and then it stabilised at 25 L/(m^2^.h) for XLE and 27 L/(m^2^.h) for ESPA at 10 bar.

In Figure 3, a relatively uniform flux was observed from 50 min to 200 min of RO operation. Also, it was very important that, after 200 min, there was no need for cleaning for both membranes. The membranes with a higher negative surface charge and greater hydrophilicity are less prone to fouling [12]. The ESPA membrane is more hydrophilic, as seen from Figure 2, and it can be expected that fouling mechanisms would affect the ESPA membrane to a lesser extent than the XLE. This claim was confirmed, as the flux of mixed process water was higher when using ESPA compared with XLE at 10 bars (Figure 3). The results are in agreement with Koo [12].

#### 3.3.1. Chemical Analyses after RO

After perlite filtration, wastewater from three streams was mixed in the volume ratio 5:1:1 from alumina washing, boehmite washing and wastewater after regeneration of boiler feed water, respectively. The ratio is based on actual wastewater generation in the company. Such a mixture represented the feed for an RO. The chemical analysis is shown in Table 4.

The concentrations of heavy metals were not problematic, because the measured values of all heavy metals were below the MAC, as seen from Table 4. Suspended solids decreased below 3 mg/L. Exceeded were concentrations of Al, Na and Cl. Organic compounds were below 30 mg/L C. Thus, in further experiments, the focus was directed towards measurement of the mentioned parameters. The XLE membrane was applied at *TMP* 7 bar. All metal concentrations remained below the MAC. Suspended solids decreased to 1.2 mg/L. Na decreased from 245 to 11 mg/L which is above 95%. In the production process Al_2_O_3_ decreases with the increase of alkali concentration [24], therefore, the concentration of Na should be low in the permeate. The measured values were acceptable. Similarly, Cl decreased to 13 mg/L, which also means 95% removal. Al decreased below MAC (3 mg/L) to 1 mg/L, which means 88% efficiency. Although the ammonia concentration was not problematic, it could be further reduced by lowering pH value below 6 [25]. The study shows ammonnia removal at 99.8%.

In the next experiment ESPA was applied at *TMP* = 7. The results were very similar to those of the XLE membrane: Na and Cl decreased by 96% and Al by 91%. Suspended solids decreased to the same value of 1.2 mg/L. The results of NaCl rejection correlated well with the reported values, around 98% for the XLE, and a little higher for the ESPA membrane at 8 bar [23].

In the third experiment the XLE membrane was tested at *TMP* 10 bar. Very similar results were obtained as at 7 bar. Only the efficiencies were a little lower compared with 7 bar for Al, Na and Cl, up to 90%.

In the fourth experiment the ESPA membrane was tested at *TMP* 10 bar. Very similar results were obtained as at 7 bar. Only the efficiencies were a little lower compared with 7 bar, for Al, Na and Cl, up to 91%.

The next experiments were performed with reverse osmosis in concentration mode with both membranes. The concentrate from the Aqua Cleer system was returned to the feed solution. The concentration factor was calculated at 5. The main advantage of such treatment is decreasing of retentate quantity production and waste streams` minimisation. The results using ESPA are presented in Figure 4. The mass concentration of Na and Cl ions increased due to the concentration of the feed. After the treatment, the concentrations of both ions decreased below the MAC. The concentration of Na was decreased from 1678 mg/l in the feed to 145 mg/l below the MAC of 200 mg/l. Due to neutralisation using HCl prior to precoat filtration and concentration of the RO feed, the concentration of chlorides in the RO feed tank was measured at 2376 mg/l. The concentration of Cl decreased to 193 mg/l, well below the MAC.

Suspended solids were removed below 2 mg/L. Al also remained below 3 mg/L.

The chemical analysis of water treated with the XLE membrane was a little worse. The membrane started to foul after concentration factor 3, therefore, the initial concentrations were lower compared with the experiment using the ESPA membrane. The concentrations of Na, and Cl decreased below the MAC, but Al remained above 3 mg/L. From the experiments we can conclude that the ESPA membrane is more appropriate for wastewater treatment and water reuse.

#### 3.3.2. Membrane Fouling Studies

The MFI was determined according to the results of Equation (3). The MFI was determined for untreated wastewater and pre-treated wastewater. The formation of a cake layer was confirmed, due to the much lower fouling index after the pre-coat filtration prior to RO treatment. The MFI of untreated water was determined at 5.7 s/L^2,^ and that of pre-treated water with a precoat filter decreased to 1.9 s/L^2^, as seen from Figure 5. The results are in agreement with suspended solids` removal as discussed in Section 3.2.

#### 3.3.3. Membrane Cleaning

In wastewater after alumina washing, different concentrations of Al, SIO_2_, Cl, and Na ions are still present, which contribute to the inorganic fouling on the membrane`s surface. Sodium phosphate species react with the deposited scales [4]. Fouling was more alleviated using the ESPA membrane, due to intensification of the inorganic fouling. The results of flux in Figure 4 show that the flux of XLE was lower than the ESPA membrane, which was probably due to stronger interactions of the inorganic compounds with the membrane compared to the ESPA. 

After 200 min of working the membrane fouled (not shown in paper), because there was no flux and cleaning was necessary. Biofouling was controlled by UV lamp in accordance with the literature [26]. The cleaning agents were among a wide variety of chemicals, including acids (HCl, HNO_3_, and H_2_SO_4_), a base (NaOH), a complexing agent (EDTA), a surfactant (SDS), and their combination. To achieve high cleaning efficiency, the effects of physical factors were studied (velocity, temperature, and time). The result showed that the two stages, caustic and detergent cleaning including NaOH-SDS followed by acid, provided an effective recovery. Since we could not disassemble the module (we had only this one), after performance of the cleaning, the effluent solution with deposits from the membrane surface was observed through SEM imaging. From Figure 6 it can be seen that small particles were washed out in a range from below 1 µm up to 20 µm. Based on the chemical analyses it was assumed that some aggregates of Al and SiO_2_ were formed_,_ as well as Na or Ca salts such as sulphates.

Cleaning with HCl probably removed the Ca and Mg salts, which are soluble in acids. This finding is in good agreement with what had been observed from the water recycling plant in another study, whereby NaOH was found to be more effective in recovering the flux than HCl [12].

The cake layer forms by deposition of material on the membrane`s surface rather than by penetration, in accordance with the SEM image.

#### 3.3.4. Process Scheme

The process scheme was designed based on the results. Figure 7 shows the scheme of the process water treatment. The water from alumina scrubbing, wastewater generated after boehmite washing and the effluents after regeneration of an ion exchanger for boiler water treatment, were gathered in a collection tank with the volume 100 m^3^. After neutralisation with 35% HCl the pre-coat filtration was performed using a pressure leaf (Kelly) filter. A powerful centrifugal pump is required for dosing the perlite precoat. The pre-treated water then flows on to the RO plant. The permeate is collected in a 60 m^3^ tank. After the treatment, the water can be reused as process water in the production process of alumina and boehmite.

In Table 5 the costs are evaluated, based on the company’s data (a) and the literature [27]. According to the results 85% of water could be reused. The cost would decrease due to water savings and reduced wastewater discharge by 82,000 Eur per year. The annual cost was calculated to be 46,630 Eur (Table 5), based on 10 years of depreciation. The payback period of less than one year indicates a high return on investment for the proposed RO wastewater treatment plant.

## 4. Conclusions

The study shows that the wastewaters could be treated in order to get quality for water reuse. The results confirmed that three flows could be collected, neutralised and treated further. After the mixture of differet wastewater types the water could be reused. The sodium content reduced by 95%, chloride from 255 down to 193 mg/L, and nitrogen below 10 mg/L. Metals such as iron and aluminium decreased below detection limits. Also, the TOC was measured below 10 mg/L. For RO the most suitable membrane was Hydranautics ESPA. It was calculated that 7 membrane modules with an area of 37.1 m^2^ each would satisfy the needs for treatment of 250 m^3^/d. The process scheme showed that we could reduce the water consumption effectively at 85% by the proposed wastewater treatment.

## Figures and Tables

**Figure 1 membranes-11-00976-f001:**
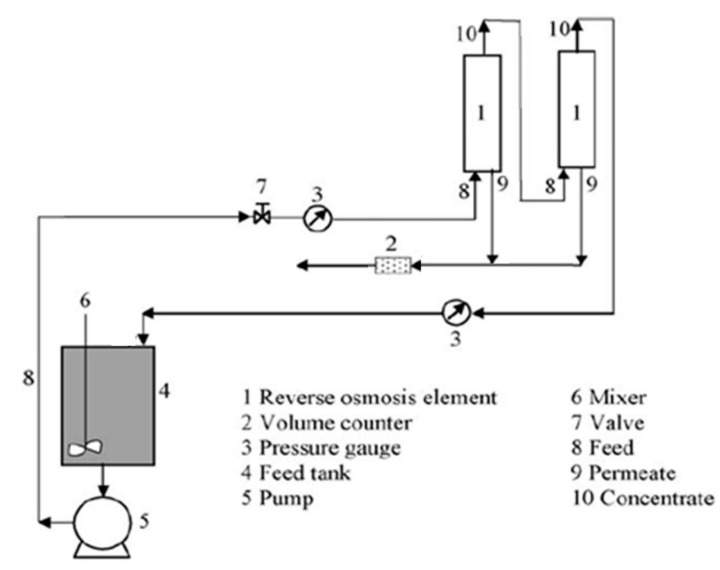
Schematic RO process.

**Figure 2 membranes-11-00976-f002:**
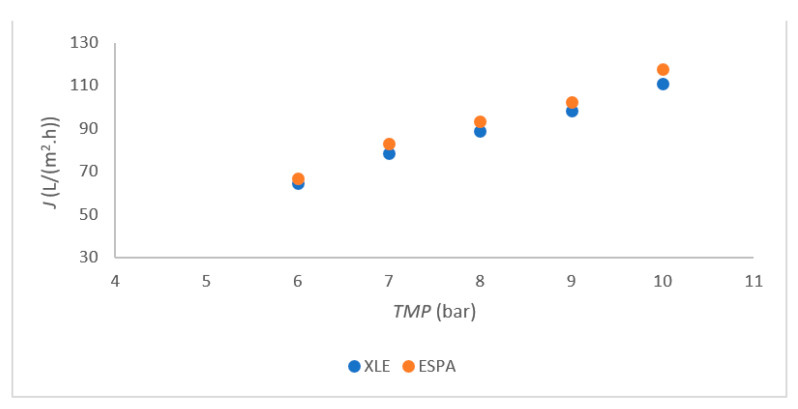
Flux in dependence of *TMP* for both membranes.

**Figure 3 membranes-11-00976-f003:**
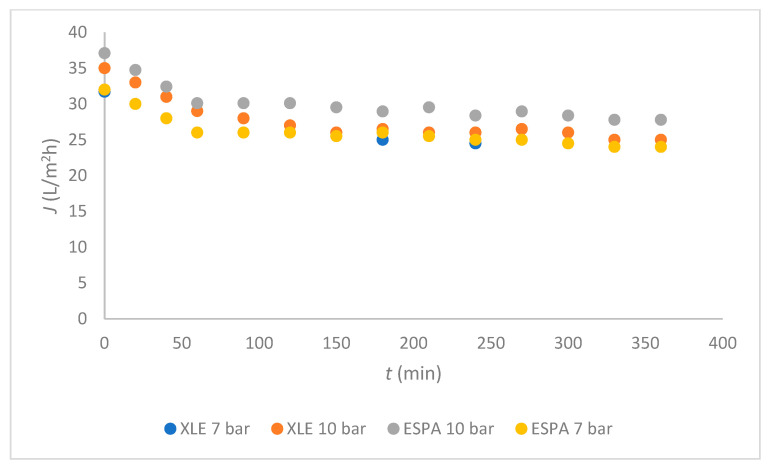
Flux dependent on time for both membranes.

**Figure 4 membranes-11-00976-f004:**
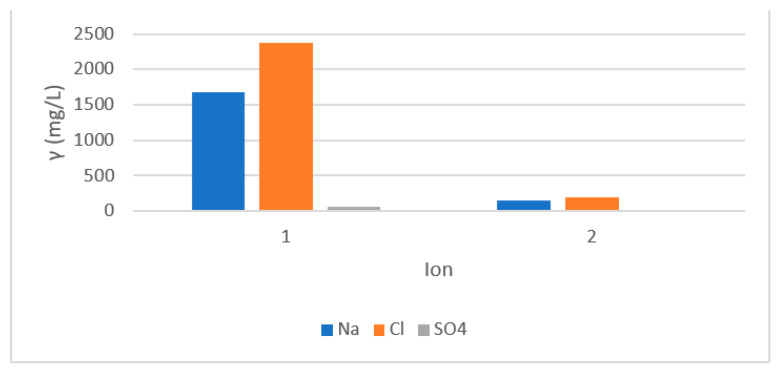
Mass concentration in feed (denoted 1) and permeate (denoted2).

**Figure 5 membranes-11-00976-f005:**
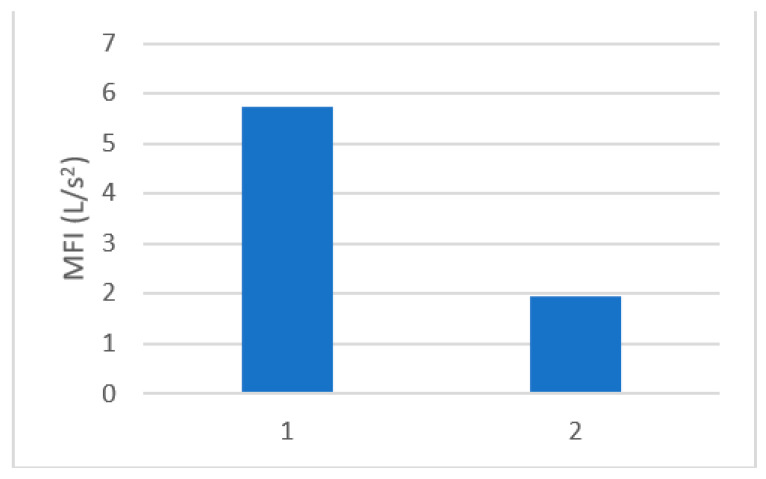
MFI for 1. Untreated mixed wastewater, 2. Pre-treated mixed wastewater.

**Figure 6 membranes-11-00976-f006:**
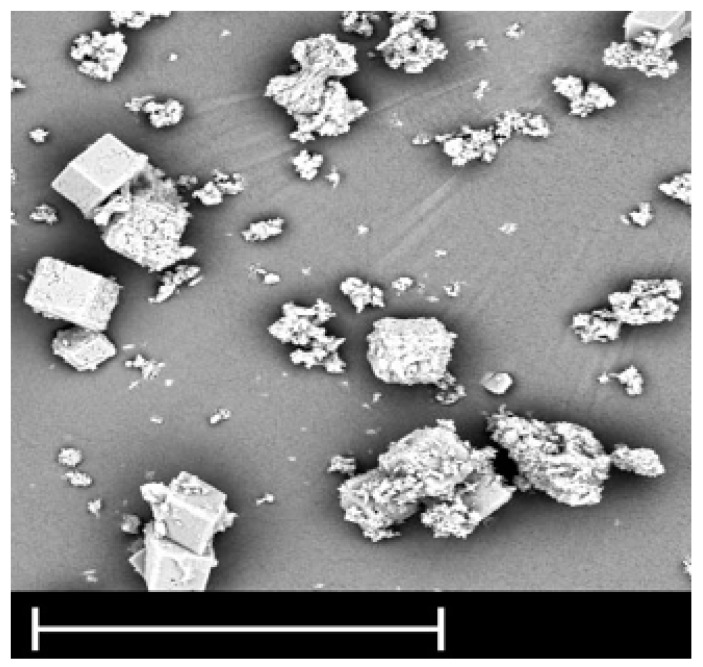
SEM image of deposit from the membrane after cleaning (line: 10 µm).

**Figure 7 membranes-11-00976-f007:**
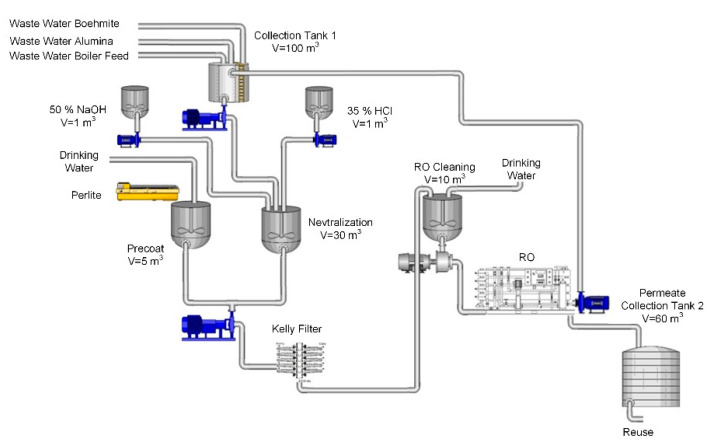
Schematic process scheme for wastewater treatment.

**Table 1 membranes-11-00976-t001:** The RO membranes’ characteristics.

Parameter	FILMTEC XLE-2521	ESPA-2521
Producer	DuPont-Filmtec (Germany)	Hydranautics, Nitto (UK)
*P* _max_	41 bar	22 bar
*T* _max_	45 °C	45 °C
*pH*	2–11	2–10
Material/Type	Spiral wound composite polyamide	Spiral wound composite polyamide

**Table 2 membranes-11-00976-t002:** The methods used for wastewater chemical analyses.

Parameter	Standard Method	Apparatus
*T* (°C)	ISO 10523	Thermometer
*pH*	ISO 10523	pH-meter, MA 5740
*A* (436 nm)	SIST EN ISO 7887	Spectrophotometer Carry 100
κ (mS/cm)	EN 27888	Conductivity-meter
*COD* (g/L O_2_)	ISO 6060	Digestion, Titration
SS	ISO 38409-H9-2	Imhoff funnel
Metals	ISO 17294-2	ICP-MS Agilent 7700x
Ions (Cl, SO_4_, NO_3_)	ISO 10304-1	IC Metrohm IC 761
Phosphorus	ISO 6878	Spectrophotometer Carry 100
Silicium dioxide	SM4500-SiO2C	Spectrophotometer Carry 100

**Table 3 membranes-11-00976-t003:** Measured chemical parameters after alumina washing.

Parameter	MAC (mg/L)	c (WW1) (mg/L)	c (WW2) (mg/L)	c (WW3) (mg/L)
Al	3	8.41	33.03	1.13
SiO_2_	250	0.74	2.86	40.1
Na	200	152	245	166
Cu	0.5	0.01	0.01	0.01
Cr	0.5	0.08	0.006	0.003
Ti	1	0.001	0.001	0.001
Fe	2	<0.1	<0.1	0.17
Zn	2	<0.01	0.02	0.14
Ni	0.5	<0.001	<0.001	0.003
Mg	10	1.7	0.1	0.3
N-NH_3_	10	21.1	0.06	0.2
N-NO_3_	20	11	9.2	9.3
SO_4_	2000	22.7	15.6	15.7
PO_4_	1	<0.05	0.08	1.03
Cl	250	21	16	203
*TOC*	30	8	9	7
*SS*	3	9	11	0.1

**Table 4 membranes-11-00976-t004:** Measured chemical parameters in the wastewater mixture (mixWW).

Parameter	MAC (mg/L)	c (mixWW) (mg/L)
Al	3	9.02
SiO_2_	250	15
Na	200	229
Cu	0.5	0.01
Cr	0.5	0.08
Ti	1	0.001
Fe	2	<0.1
Zn	2	<0.01
Ni	0.5	<0.001
Mg	10	1.6
N-NH_3_	10	1.1
N-NO_3_	20	10
SO_4_	2000	24
PO_4_	1	<0.05
Cl	250	255
*TOC*	30	8
*SS*	3	2

**Table 5 membranes-11-00976-t005:** Cost estimation.

	Total Cost (Eur)	Annual Costs/10 y Depreciation (Eur)
Cost of plant [27]	180,000	18,000
Energy ^a^	7730	7730
Chemicals [27]	3900	3900
Labour ^a^	11,000	11,000
Wastewater discharge ^a^	6000	6000
Total	208,630	46,630

^a^ Compay’s data.

## Data Availability

Data supporting the reported results can be found here: https://dk.um.si/Dokument.php?id=145897 (accessed on 2 December 2021).

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
