# Peer review of "Reverse Osmosis Treatment of Wastewater for Reuse as Process Water—A Case Study"

_membranes, 2021, doi:10.3390/membranes11120976_

Round 1

Reviewer 1 Report

The paper titled “Reverse osmosis treatment of wastewater for reuse as process water – A case study” and written by Marjana Simonič is interesting and reports results about RO technology applied to a specific wastewater concerning boiler processes. The introduction and the number of references should be extended. I recommend a major revision based on the following comments:

  1. The title is very general; it should be more specific. I mean, this work is about wastewater in boiler process right? Not for all kind of wastewater.
  2. A recommendation for one of the keywords. I would write “wastewater reuse” instead of “reuse”.
  3. Page 1, line 45. Please, write “reverse osmosis (RO) membranes” and use the abbreviation in the rest of the document. For example, in Page 2 lines 52,56, 69. Title of section 3.3, 3.3.1, in page 7 line 243, page 8 line 272, page 11 lines 359, 375.
  4. This study is about applying RO technology to a wastewater to be reuse. The third paragraph of the introduction is about RO technology but, it is short. This part should be extended as the topics and number of publications about RO is important. The author mentioned the design of spacer for specific applications but, they should also mention that the permeability coefficients with the feed spacer design has a relevant impact on the performance of RO membranes in terms of production, permeate quality and specific energy consumption. This is due to relation between feed spacer geometry and pressure drop and concentration polarization phenomena.
  5. The author also mentioned that RO system are easy to operate. Author is right as RO technology is quite reliable and well established. It also should be mentioned that the operation of RO system deals with the impact of fouling, which is the main concern of this technology and the main responsible for loosing performance and efficiency in long term-operation.
  6. Scaling in RO could be a serious problem depending in the inorganic composition of the feedwater and the recovery rate of the RO system. In fact, scaling is on of the limiting factors for increasing the flux recovery rate and improve the efficiency of the process. This issue should be documented.
  7. I think the following study about pretreatments of RO technology should be included:
    1. Reverse osmosis pretreatment technologies and future trends: A comprehensive review
  8. Author did not mention the importance of optimal design and finding optimal operation points in RO systems and how it can affect the performance and viability of the process.
  9. Page 2, line 92. Please, define TMP as transmembrane pressure (TMP). If it is a variable write in italics in the entire document.
  10. Page 2, line 87. There is a space in 72000. Same page, line 88, 3 should be written as superscript.
  11. Page 3, line 117. COD was already abbreviated in a previous page.
  12. Table 3, parameters should not be written in italics as they are chemical compounds. TOC is ok.
  13. There are some relevant papers related with RO for reusing wastewater that should be included at least in a general way.

Author Response

Responses are written below comment:

  1. The title is very general; it should be more specific. I mean, this work is about wastewater in boiler process right? Not for all kind of wastewater.

Response:

The work is not only about wastewater in boiler process, but also wastewater after alumina and boehmite production (please see the last paragraph in Introduction section). So three types of water are gathered and cleaned, therefore I believe that the Title is suitable.

  1. A recommendation for one of the keywords. I would write “wastewater reuse” instead of “reuse”.

Response: The keyword was added as suggested.

  1. Page 1, line 45. Please, write “reverse osmosis (RO) membranes” and use the abbreviation in the rest of the document. For example, in Page 2 lines 52,56, 69. Title of section 3.3, 3.3.1, in page 7 line 243, page 8 line 272, page 11 lines 359, 375.

Response: Thank You for the comment, it was done as sugessted and checked throughout the text.

  1. This study is about applying RO technology to a wastewater to be reuse. The third paragraph of the introduction is about RO technology but, it is short. This part should be extended as the topics and number of publications about RO is important. The author mentioned the design of spacer for specific applications but, they should also mention that the permeability coefficients with the feed spacer design has a relevant impact on the performance of RO membranes in terms of production, permeate quality and specific energy consumption. This is due to relation between feed spacer geometry and pressure drop and concentration polarization phenomena.

Response: The description was added into the article, please see P2L49-52.

  1. The author also mentioned that RO system are easy to operate. Author is right as RO technology is quite reliable and well established. It also should be mentioned that the operation of RO system deals with the impact of fouling, which is the main concern of this technology and the main responsible for loosing performance and efficiency in long term-operation.

Response: The description was added into the article, please see P2L54-56.

  1. Scaling in RO could be a serious problem depending in the inorganic composition of the feedwater and the recovery rate of the RO system. In fact, scaling is on of the limiting factors for increasing the flux recovery rate and improve the efficiency of the process. This issue should be documented.

Response: The description was added in P2L62-65.

  1. I think the following study about pretreatments of RO technology should be included:
    1. Reverse osmosis pretreatment technologies and future trends: A comprehensive review

Response: The reference was included: [14].

  1. Author did not mention the importance of optimal design and finding optimal operation points in RO systems and how it can affect the performance and viability of the process.

Response: Brief explanation of importance of optimal design and finding optimal operation points in RO systems could be find in first paragraph on page 2.

  1. Page 2, line 92. Please, define TMP as transmembrane pressure (TMP). If it is a variable write in italics in the entire document.

Response: It was corrected as suggested.

  1. Page 2, line 87. There is a space in 72000. Same page, line 88, 3 should be written as superscript.

Response: It was corrected as suggested.

  1. Page 3, line 117. COD was already abbreviated in a previous page.

Response: It was corrected as suggested; only COD was written in P3L117.

  1. Table 3, parameters should not be written in italics as they are chemical compounds. TOC is ok.

Response: It was corrected as suggested.

  1. There are some relevant papers related with RO for reusing wastewater that should be included at least in a general way.

Response: Some references were included as suggested [6,11,12,13,14].

Reviewer 2 Report

Please remake the figure 1., because in your Figure all the permeate and the concentrate are returned to the feeding tank, making the process non useful!

Remake the part bellow, because table 1 and table 2 are in fact Table 3. Also use WW1, WW2 and WW3 for all the waste water to be more clear.

After filtration of all samples, parameters listed in Table 1 were measured in  wastewater samples. Wastewater was taken from the washing process line for 10 times  and mixed. Such average sample was analysed for all parameters according to the Slovenian regulation. Washing water in alumina production is denoted in Table 1 as WW1, washing water in boehmite production is denoted in Table 1 as WW2 and water after  regeneration during feed boiling water condition as WW3. The results are gathered in Table 2. Al exceeded 3 mg/L in WW1. The suspended solids mass concentration was too  high in washing water in boehmite production, sodium was above MAC and aluminium  was very high measured at 33 mg/L. In WW3 phosphorus was just above MAC and pH value was too alkaline.

Remake tables 3 and 4 because there are missing the suspended solids.

 Correct the Figure caption,

Figure 4. Mass concentration if feed (denoted 1) and permeate (denoted2)

with

Figure 4. Mass concentration in feed (denoted 1) and permeate (denoted2)

The Figure 8 must be remade: I do not see the RO device. I do not see where do you receive the concentrate. Why do you recycle the permeate in the feeding?

The article needs moderate English correction.

Author Response

Responses are written below comment:

Please remake the figure 1., because in your Figure all the permeate and the concentrate are returned to the feeding tank, making the process non useful!

Response: The Figure 1 was remade, so that permeate is collected/withdrawn and concentrate is returned to the feed.

Remake the part bellow, because table 1 and table 2 are in fact Table 3. Also use WW1, WW2 and WW3 for all the waste water to be more clear.

After filtration of all samples, parameters listed in Table 1 were measured in  wastewater samples. Wastewater was taken from the washing process line for 10 times  and mixed. Such average sample was analysed for all parameters according to the Slovenian regulation. Washing water in alumina production is denoted in Table 1 as WW1, washing water in boehmite production is denoted in Table 1 as WW2 and water after  regeneration during feed boiling water condition as WW3. The results are gathered in Table 2. Al exceeded 3 mg/L in WW1. The suspended solids mass concentration was too  high in washing water in boehmite production, sodium was above MAC and aluminium  was very high measured at 33 mg/L. In WW3 phosphorus was just above MAC and pH value was too alkaline.

Response:

The reviewer is right and I have remake the paragraph as suggested (please see chapter 3.1).

Remake tables 3 and 4 because there are missing the suspended solids.

Response: The reviewer is right and I have remade the paragraph by adding SS as suggested.

 Correct the Figure caption,

Figure 4. Mass concentration if feed (denoted 1) and permeate (denoted2)

with

Figure 4. Mass concentration in feed (denoted 1) and permeate (denoted2)

Response:

I have corrected the Figure 4 caption as suggested.

The Figure 8 must be remade: I do not see the RO device. I do not see where do you receive the concentrate. Why do you recycle the permeate in the feeding?

Response: The Figure 8 was remade, so that the concentrate receiving and permeate withdrawal is seen.

The article needs moderate English correction.

 Response: English correction was made by native speaker.

Reviewer 3 Report

Reverse osmosis treatment of wastewater for reuse as process water- A case study

The goal of this manuscript was to purify wastewater from an alumina washing production washing line, a boehmite production line, and an ion exchange regenerant to surface water discharge quality. Overall, there is merit to this work, however, I have several comments that should be addressed prior to publication.

Specific comments below:

Introduction:

The author is a bit unclear in both the abstract and the introduction in terms of what wastewaters were used. Based on my reading, I was expecting the author to test each of the three types of wastewater separately. However, based on my reading of the materials and methods, these waters were mixed. The author should rephrase the introduction to clarify this.

Materials and Methods:

Wastewater is typically disinfected during reuse upstream of reverse osmosis membranes to prevent biofouling in RO units. Was the water disinfected upstream? If not, author should note this, and potentially discuss in the results and discussion section

Lines 90-97. I am curious what the water recovery is through the RO system?

Results and Discussion:

Section 3.2: Please rephrase the HCl addition, right now it can be read as “the wastewater mixture was 35% HCl”

Section 3.3.1: The removal of ammonia by the RO system seems low. Please consider the following reference to help contextualize the results: DOI: 10.1039/D0EW01112F (Paper) Environ. Sci.: Water Res. Technol., 2021, 7, 739-747

Line 281: How much acid was added to the system? A value of 2376 mg/L of chloride seems very high given that the chloride concentration in all of the wastewaters is below 25 mg/L (as stated in Table 3). Is adding this much acid feasible in practice?

Line 321: Based on Figure 3, there is no difference in membrane flux between the 50 min and 350 min of operation. Why does the author say that the membrane was fouled at 200 min? What indicates this?

Line 371: What indicates that the wastewaters can be treated together vs. separately? The author did not measure the wastewaters separately. Please rephrase.

Conclusions:

How much would the water consumption be reduced by? I would further consider the cost of the treatment system.

Author Response

Response is written below specific comment:

Introduction:

The author is a bit unclear in both the abstract and the introduction in terms of what wastewaters were used. Based on my reading, I was expecting the author to test each of the three types of wastewater separately. However, based on my reading of the materials and methods, these waters were mixed. The author should rephrase the introduction to clarify this.

Response: The Abstract and Intorduction were rephrased and I hope that the aim is now more clearly expressed. The wastewaters were mixed in accordance with the requirements of the company.

Materials and Methods:

Wastewater is typically disinfected during reuse upstream of reverse osmosis membranes to prevent biofouling in RO units. Was the water disinfected upstream? If not, author should note this, and potentially discuss in the results and discussion section

 Response:

The water was disinfected upstream by UV lamps.

Lines 90-97. I am curious what the water recovery is through the RO system?

 Response:

The water recovery is estimated at 80 % in accordance with other reported literature (at concentration factor 5).

Results and Discussion:

Section 3.2: Please rephrase the HCl addition, right now it can be read as “the wastewater mixture was 35% HCl”

 Response:

The addition of 35 % HCl was rephrased.

Section 3.3.1: The removal of ammonia by the RO system seems low. Please consider the following reference to help contextualize the results: DOI: 10.1039/D0EW01112F (Paper) Environ. Sci.: Water Res. Technol., 2021, 7, 739-747

Response: The reference was carefully read, however the system consisted of bioreactor which improved the ammonia removal in proposed reference. In my paper, there was no anaerobic pretreatment, therefore the concentration remained higher.

Line 281: How much acid was added to the system? A value of 2376 mg/L of chloride seems very high given that the chloride concentration in all of the wastewaters is below 25 mg/L (as stated in Table 3). Is adding this much acid feasible in practice?

Response: Your observation is correct, the value of Cl concentration is high. We did not measure the volume of HCl, however I have checked the value for chloride in waste water in boiler system and somehow the 0 disappeared, the correct value is 203 mg/L Cl-. I have corrected the value in Table 3. As the concentrate returns to the feed, the value of chlorides increases.

 Line 321: Based on Figure 3, there is no difference in membrane flux between the 50 min and 350 min of operation. Why does the author say that the membrane was fouled at 200 min? What indicates this?

Response: In Figure 3 the flux without concentrate recycle is shown. If the concentrate was recycled back to the feed, the flow at 200 min was parctically close to zero (not shown in paper) and therefore we stopped the system and perform cleaning.

Line 371: What indicates that the wastewaters can be treated together vs. separately? The author did not measure the wastewaters separately. Please rephrase.

Response: The sentence was rephrased. Chemical analyses showed that wastewater quality did not differ much, therefore with the company representative we decided to treat three types of wastewatr together.

Conclusions:

How much would the water consumption be reduced by? I would further consider the cost of the treatment system.

Response: The water consumption would be reduced by about 90 %.

Round 2

Reviewer 1 Report

The author has addressed all my comments

Author Response

The author has addressed all my comments.

Response: Thank You.

Reviewer 2 Report

English was much improved

Revision is still needed:

Change:

The results are gathered in Table 4. Al exceeded 3 mg/L in WW1. The suspended solids` mass concentration was too high in the washing water in boehmite production, sodium was above the MAC and aluminum was very high, measured at 33 mg/L.

With:

The results are gathered in Table 3. Al exceeded 3 mg/L in WW1. The suspended solids` mass concentration was too high in the washing water in boehmite production (WW2), sodium was above the MAC and aluminum was very high, measured at 33 mg/L.

Change:

The mixture of wastewater streams was neutralised to pH = 7 using HCl (w=35 %).

With:

The mixture of wastewater streams was neutralised to pH = 7 using HCl 35 % (weight percents)..

Figure 8 must be changed:

Under the permeate tank you are writing Recycling , which is wrong. You should put the pump for recycling after the RO  on the concentrate line recycling.

Author Response

Rew2

Revision is still needed:

Change:

The results are gathered in Table 4. Al exceeded 3 mg/L in WW1. The suspended solids` mass concentration was too high in the washing water in boehmite production, sodium was above the MAC and aluminum was very high, measured at 33 mg/L.

With:

The results are gathered in Table 3. Al exceeded 3 mg/L in WW1. The suspended solids` mass concentration was too high in the washing water in boehmite production (WW2), sodium was above the MAC and aluminum was very high, measured at 33 mg/L.

Response: The correction was made as suggested (Please see L175-178)

Change:

The mixture of wastewater streams was neutralised to pH = 7 using HCl (w=35 %).

With:

The mixture of wastewater streams was neutralised to pH = 7 using HCl 35 % (weight percents)..

Response: The correction was made as suggested.

Figure 8 must be changed:

Under the permeate tank you are writing Recycling , which is wrong. You should put the pump for recycling after the RO  on the concentrate line recycling.

Response: By »Recycling« It was ment reuse as process water in industrial process. However, the correction was made as suggested.

Reviewer 3 Report

I thank the author for the responses to the comments.

There are minor comments I would like to address in response to the author’s response to the comments:

Initial comment: Wastewater is typically disinfected during reuse upstream of reverse osmosis membranes to prevent biofouling in RO units. Was the water disinfected upstream? If not, author should note this, and potentially discuss in the results and discussion section

Authors  Response: The water was disinfected upstream by UV lamps.

Follow up: UV disinfection by itself does not provide a disinfectant residual and is therefore not appropriate for controlling biofouling in RO units. Please include a discussion in the results and discussion section.

Initial comment: Section 3.3.1: The removal of ammonia by the RO system seems low. Please consider the following reference to help contextualize the results: DOI: 10.1039/D0EW01112F (Paper) Environ. Sci.: Water Res. Technol., 2021, 7, 739-747

Authors Response: The reference was carefully read, however the system consisted of bioreactor which improved the ammonia removal in proposed reference. In my paper, there was no anaerobic pretreatment, therefore the concentration remained higher.

Follow up: Anaerobic bioreactors would not reduce ammonia concentrations in sewage and therefore should not affect ammonia removal efficiency by RO membranes. Hence, it should be relevant to the results presented here.

 Line 321: Based on Figure 3, there is no difference in membrane flux between the 50 min and 350 min of operation. Why does the author say that the membrane was fouled at 200 min? What indicates this?

Authors Response: In Figure 3 the flux without concentrate recycle is shown. If the concentrate was recycled back to the feed, the flow at 200 min was parctically close to zero (not shown in paper) and therefore we stopped the system and perform cleaning.

Follow up: Please clarify this in the manuscript.

Initial comment: Conclusions: How much would the water consumption be reduced by? I would further consider the cost of the treatment system.

Authors Response: The water consumption would be reduced by about 90 %.

Follow up: Great! Please consider including this in the manuscript, and please consider discussing the cost of the treatment system.

Author Response

There are minor comments I would like to address in response to the author’s response to the comments:

Initial comment: Wastewater is typically disinfected during reuse upstream of reverse osmosis membranes to prevent biofouling in RO units. Was the water disinfected upstream? If not, author should note this, and potentially discuss in the results and discussion section

Authors  Response: The water was disinfected upstream by UV lamps.

Follow up: UV disinfection by itself does not provide a disinfectant residual and is therefore not appropriate for controlling biofouling in RO units. Please include a discussion in the results and discussion section.

Response: The reviewer is correct that UV lamp do not have residual. I have to emphasize that the quality of filtered water after RO was such that the UV seamed to be appropriate for controling biofouling. The reference was included which confirmed our assumption: 10.5004/dwt.2011.2377 The discussion was added, please see line 240-242.

Initial comment: Section 3.3.1: The removal of ammonia by the RO system seems low. Please consider the following reference to help contextualize the results: DOI: 10.1039/D0EW01112F (Paper) Environ. Sci.: Water Res. Technol., 2021, 7, 739-747

Authors Response: The reference was carefully read, however the system consisted of bioreactor which improved the ammonia removal in proposed reference. In my paper, there was no anaerobic pretreatment, therefore the concentration remained higher.

Follow up: Anaerobic bioreactors would not reduce ammonia concentrations in sewage and therefore should not affect ammonia removal efficiency by RO membranes. Hence, it should be relevant to the results presented here.

Response: Anaerobic indeed would not reduce ammonia concentration, I was a bit unclear. I would like to emphasize that the SAF-MBR system would reduce ammonia concentration. In mentioned reference it was stated:« At pH 6, the TAN rejection efficiency was optimal at 99.8%. At a pH > 6, the passage of uncharged NH3 increased, decreasing TAN removal.« In our case pH was above 7 and consequently the removal was lower. However, I have added the reference and discussion, please see: L272-274.

 Line 321: Based on Figure 3, there is no difference in membrane flux between the 50 min and 350 min of operation. Why does the author say that the membrane was fouled at 200 min? What indicates this?

Authors Response: In Figure 3 the flux without concentrate recycle is shown. If the concentrate was recycled back to the feed, the flow at 200 min was parctically close to zero (not shown in paper) and therefore we stopped the system and perform cleaning.

Follow up: Please clarify this in the manuscript.

 Response: Done, please see L340-342.

Initial comment: Conclusions: How much would the water consumption be reduced by? I would further consider the cost of the treatment system.

Authors Response: The water consumption would be reduced by about 90 %.

Follow up: Great! Please consider including this in the manuscript, and please consider discussing the cost of the treatment system.

Response: I agree with Reviewer comment, however the costs of treatment system was not the aim. Now, I have included the paragraph discussing the cost (please see below Figure 8):

 In Table below the costs are gathered, based on data of company (c) and literature [27]

total (Eur)

annual/10 y depreciation

Cost of plant [27]

180,000

18,000

Energyc [27]

7,730

7,730

Chemicals [27]

3,900

3,900

Labourc

11,000

11,000

Wastewater dischargec

6,000

6,000

Total

208,630

46,630

[27] https://doi.org/10.1016/j.desal.2017.05.038

The cost due to water saving and lower wastewater discharge would be reduced for 82,000 Eur per year. The  annual cost for RO treatment system is calculated at 46,630 Eur. The payback period of less than one year indicates a high return on investment for the proposed RO wastewater treatment plant.